# Social context in political stance detection: Impact and extrapolation

**Ramon Villa-Cox**[1,2]*, **Evan M. Williams**[2], **Kathleen M. Carley**[2]

**1** ESPAE, Escuela Superior Politecnica del Litoral, Guayaquil, Guayas, Ecuador, **2** Software and Societal Systems, Carnegie Mellon University, Pittsburgh, Pennsylvania, United States of America

* ramavill@espol.edu.ec

## Abstract

Stance detection is an important task with a wide range of high-impact social applications, including opinion polling and detecting propaganda, misinformation, and hate speech. In this work, we explore the performance and extrapolation power of political stance-detection models using an existing large-scale weakly-labeled Twitter dataset collected around the 2019 South American Protests. We construct transformer-based user and tweet encoders to embed users in a low-dimensional space using their posts and interactions. We then train heterogeneous graph attention networks to predict user stances and contrast their ability to extrapolate stance predictions to different country contexts and to future events. We find that leveraging users' ego-network in political stance detection improves in-country model performance for every country we examine. More notably, we find that leveraging a user's social context greatly enhances the ability of our stance detection models to extrapolate to new country contexts and future data.

## 1 Introduction

Public opinion toward governments and their policies has widespread social implications. Public support is often viewed as a kind of currency in political science: more public support can allow governments more freedom in their actions, whereas widespread public disapproval can constrain the set of actions that a government can take. Studies have found public opinion to be a strong predictor of policy [1,2], and information about public opinion can impact the decisions and planning of politicians, businesses, foreign governments, and people. It is therefore unsurprising that global election and opinion polling has grown into a 6.78 billion dollar industry [3]. However, opinion polling often lags behind real events, and in unstable political situations, this can engender widespread uncertainty. Surveys, especially in dynamic or unstable political situations, are constricted by resources, and the diversity of participants is confined by the survey method. To mitigate these constraints, researchers have explored automated opinion mining, a stance detection sub-task, in social media settings.

In some cases, large-scale social media data has served as an alternative way to gauge public opinion [4]. However, it has long been recognized in the stance detection field that considering only an isolated sentence will provide an incomplete assessment of a user's stance towards a predefined target [5]. Despite this, little research has been done to integrate the

**Data availability statement:** In compliance with Twitter's (now known as X) 2023 Terms of Service, we are not able to share tweet or user ids, nor the corresponding text for each post.

However, to ensure reproducibility of the results presented in the paper, we make available all predictions obtained for the different models for anonymized users and them corresponding anonymized tweets. This can be used to recreate the country level results and the different robustness analysis. The code developed is publicly available at: ● https://github.com/rvillaco/Protest_Stance_Detection ● https://doi.org/10.5281/zenodo.15571840 The anonymized data, including the User Embeddings, required to reproduce the results presented in the paper is available at: ● https://doi.org/10.5281/zenodo.14207926.

**Funding:** The research for this paper was supported in part by the ARMY Scalable Technologies for Social Cybersecurity, Office of Naval Research, MURI: Persuasion, Identity, & Morality in Social-Cyber Environments, and Office of Naval Research, MURI: Near Real Time Assessment of Emergent Complex Systems of Confederates under grants W911NF20D0002, N000142112749, and N000141712675. RVC was also supported by the Secretaría de Educación Superior, Ciencia, Tecnología e Innovación (https://siau.senescyt.gob.ec/convocatorias), Ecuador; the Center for Informed Democracy and Social-cybersecurity (https://www.cmu.edu/ideas-social-cybersecurity) and the Center for Computational Analysis of Social and Organizational Systems (CASOS) at Carnegie Mellon University. The views and conclusions are those of the authors and should not be interpreted as representing the official policies, either expressed or implied, of the ARMY, the ONR, or the US or Ecuadorian Government. None of the sponsors or funders of this work played any role in the study design, data collection and analysis, decision to publish, or preparation of the manuscript.

**Competing interests:** The authors have declared that no competing interests exist.

different levels of contextual user information available in social media data into stance detection models. More importantly, there is a need to evaluate the impact that this additional context can have on final classification performance [6].

In this work, we explore the in-sample performance and generalizability of country-level models trained at the various levels of contextual information available as users interact through social media. We focus on three context levels that increasingly convey more information about a given user, namely, the document/post level, the user level, and the social network level. For this task, we use a weakly-labeled large-scale Twitter (now X) dataset, that encompasses the widespread 2019 protests that erupted in Ecuador, Chile, Bolivia, and Colombia [7]. The regional nature of the dataset presents the opportunity to test the generalization capabilities of our proposed models in out-of-sample country data at different levels of contextual information. The different cultural and ideological motivations for the protests provide relevant roadblocks to test the robustness of the proposed estimators. However, given that these protests occurred concurrently, the dataset does not facilitate the testing of robustness over time. This is an outstanding problem in social media settings, as relevant textual features can change significantly in a short period. To address this, we constructed an additional weakly-labeled dataset around the 2020 Chilean Referendum. In summation, the contributions of this paper are the following:

- We propose a compartmentalized stance-detection architecture using transformers and graph neural networks that leverage a user's ego-network, a user's tweet timelines, and weakly-labeled protest-related tweets to predict their stance towards the government of each country. This architecture, albeit computationally expensive, allows us to evaluate the effect, both on in-sample performance and generalizability, of making more contextual information available to a classifier.
- Through ablation studies, we examine the cross-country performance of each stance model in a one-shot prediction setting. We found that increasing the contextual information available to a model (as defined by each compartment of the architecture) not only improved the performance of a classifier within the country it was trained but also made it more robust to out-of-sample predictions.
- We collect a novel weakly-labeled Chilean referendum dataset to explore the ability of each Chilean model, trained at different context levels, to generalize to future data. As before, we found that increasing context improves the model's generalization for future posts of users seen during the Chilean protests and for users exclusively seen in the referendum collection. We find that the inclusion of a user's heterogeneous ego-network information, by differentiating between replies and retweets, yields much larger improvements in related country-context assessments or its robustness over time.
- We make publicly available a variant of a BERT language model [8], we call *twBETO*, developed for this work and trained on a substantial corpus of 150 million Spanish tweets. This model is not only trained on more tweets than other Spanish BERT models for Twitter [9] but has better coverage of non-European Spanish dialects.

To the best of our knowledge, the evaluation of the effects of context on the generalization capabilities of these models has not been explored in other work and has been identified as an outstanding issue in the task of stance classification [6]. Moreover, even though the subject of interest of our study is political stances in South American politics during charged political events, the method proposed can nevertheless be used in understanding social change and political movements in different cultures and continents. Finally, while most user-level stance detection applications are generally benign or beneficial, e.g., targeted advertising, they

certainly have the potential to be abused by malign governments or corporations in targeting vulnerable or dissident groups. Our findings underscore the importance of anonymity and privacy on social media platforms, and remaining cognizant of one's digital footprint.

## 2 Related work

Stance detection is an essential component of many tasks associated with online social network moderation and analysis, including opinion polling and the detection of propaganda, misinformation, and hate speech. In some domains, sentiment analysis may be a reasonable approximation of stance, but it has been shown that, on Twitter, sentiment polarity is a poor proxy, particularly around contentious political issues [10]. Posts with a positive sentiment score can be used to oppose a topic and vice versa. As such, the overall sentiment of the text is not necessarily relevant to the classification [11]. Work in this area has concentrated on exploring stance in conversations (also known as rumor stance classification [12]) and on debates with respect to a predefined topic or target (known as target-stance classification). However, progress in this area has been limited by its reliance on small datasets, primarily created around challenge competitions like SemEval-2016 [10] through crowd-sourced human annotation. The work undertaken in this paper is an instance of target-stance classification. In what remains of this section we provide a brief overview of the main research that has been applied in the relevant areas. For a more detailed description of these areas and the main datasets available, we refer the reader to other comprehensive reviews of the state of the art in the subject [6].

### Target Stance Classification

Target Stance Classification focuses on classifying the stance of a user or document for a predefined topic or target. Task 6 of Semeval 2016 is a common benchmark for target stance classification. Authors have achieved SOTA results on this benchmark using architectures ranging from end-to-end neural ensemble models [13] to hand-crafted feature-based classifiers [14]. The latter relies on domain knowledge to determine the most informative features for a given problem. Due to the limited amount of data available, algorithms that rely on hand-crafted features are still prominent and achieve competitive performances with Deep Learning algorithms. Additionally, the vast majority of stance-detection literature focuses on classifying documents. Considering users adds additional complexity to the task as users can post many unrelated documents or express contradictory or nuanced stances towards targets that may not be captured in a single post. Even though there exists work that has explored user-level stance detection on Twitter data [15], the proposed approach requires pairwise similarity calculations for all users of interest, which limits the scalability of the approach to large social media networks like those we use in this work. To make the approach tractable, the authors dropped all but the most active 5k users—the pairwise similarity bottleneck limits the ability of the approach to scale to millions of users.

### GNNs for Stance Detection

Graph Neural Networks (GNNs) have become increasingly popular in recent years, but to our knowledge, have only recently been used for stance detection at the user level. Graph Neural Networks have been used in tasks related to stance in conversations at a document level, including several papers that explore GNNs for fake news classification and rumor detection. For example, a gated graph neural network model, PCGNN [16], was proposed for rumor detection on Tweet threads achieving SoTA results on the PHEME dataset [17].

Reference [18] propose a modified label propagation algorithm that they benchmark against graph neural network architectures for user-level stance prediction. Reference [19] proposed an unsupervised BERT training approach that leverages user retweet interactions and fine-tunes this for user political stance detection. Most works focus on text-level classification. While this is effective for fine-grained stance detection, no strong inference can be made about the alignment or intentions of the user who posted the false or misleading content. To combat misinformation at scale in an environment with state-backed propaganda campaigns, troll farms, and bot networks, stance detection at the user level is essential for moderation at scale.

## Generalizing Stance classification

Several recent efforts have been made to study the generalization capabilities of stance classifiers to different domains, contexts, or targets. One vein of this research, known as cross-target stance detection, focuses on applying a target stance classifier trained on one domain to a different, but related, domain. Work in this area has explored how to leverage labeled instances from related political topics to increase generalizability across targets [20,21], or how effectively stance predictions transfer across languages [22]. Similarly, other work has explored how synthetic data, generated from related targets, can improve cross-target stance detection on Tweets [23]. However, in contrast to our work, research in this field improves the generalizability of classifiers across different targets by leveraging their semantic relationship at a document level. For this reason, each of these works explores stance at the tweet/ document level, rather than at a user level. More recent work [24], proposes a methodology for multi-modal cross-target stance detection on 3,871 Twitter users, however, their model uses followership and friendship networks for each user, which are often not feasible to extract on large networks given API limitations. Importantly, our work differs from these approaches as, for a fixed target, we explore how varying the amount of contextual information for a given user (an individual post, all a user's posts, or their local ego-network) improves within-sample and out-of-sample predictions.

A similar vein of research, known as context-sensitive classification, models context in the form of spatial and temporal locality and is crucial to many tasks, including search engines, and context-aware Web applications [25]. This is highly relevant for stance detection in online social media where users engage with one another in a variety of ways, and thus focusing on an isolated sentence will likely result in an incomplete assessment of a user's stance, as they can employ humor, satire, ask a question, or talk about something else entirely [5]. Although some works have implicitly demonstrated the benefits of this approach by incorporating different levels of context in their architecture [26,27], there is a need to systematically quantify the resulting gains in classifier performance as different levels of contextual information become available to the model. This line of questioning has been identified as an outstanding issue in the task of stance classification [6], and is something this work attempts to explore.

## 3 Data description

We ground our analysis in the wave of protests that effectively paralyzed the South American region at the end of 2019. The protests in Ecuador, Chile, and Colombia were led by left-wing movements that sought to resist austerity measures, the Bolivian protests were a right-wing response to an alleged electoral fraud undertaken in favor of the president who was seeking reelection. We use a dataset that contains the stances of hundreds of thousands of Twitter users towards their respective governments during the event [7]. The labels were constructed

using a weak-labeling methodology that leveraged the users' endorsement of hand-labeled political figures as well as their usage of stance-tags (hashtag campaigns with well-defined stances towards each government and that occur at the end of a tweet). The methodology relies on the hypothesis that users are more likely to tweet (or retweet) stance-tags or political figures that are aligned with their stances during these events. For this reason, stance labels are assigned to a user if the percentage of tweets with a consistent stance-tag or retweets from political figures with a consistent stance is above a given threshold (the authors use 90% as a threshold based on a labeled validation set). A final stance label is assigned to a user if the stance obtained by both signals (usage of hashtags and retweet of political figures) is consistent. In Table 1 we provide some examples of the most prominent stance-tags observed for each country collection.

The authors validate the quality of the weakly-annotated labels based on a hand-labeled sample of users and by showing that the constructed labels partition the users in communities that are polarized in their language and news-sharing behavior in a way consistent with the ideological underpinnings of each protest. By leveraging an established machine translation algorithm [28], they show that terms related to left-leaning ideologies in one community tend to be discussed in similar contexts as right-leaning terms (e.g., Socialism mistranslates to Fascism); terms related to law and order in one group are discussed in a similar context as the other discusses oppression, or that opposition leaders are discussed in similar contexts as government representatives. However, they show that, while some ideological consistencies are maintained when comparing protest movements from different countries, differences in dialects start to dominate the polarization measures. For more details on the polarization dimensions identified in the data, refer to [7]. These semantic regularities suggest that text-based features should be informative for political stance classification, but this information should degrade when applied to different country contexts. We explore this formally in this work.

The protest dataset, which was collected between September 25 and December 24 of 2019, contains 550k labeled users split unevenly across the four countries and contains over 36 million labeled tweets. It contains an additional 1.1 million unlabeled neighbors and 40 million unlabeled tweets. Stances are imbalanced in 3 of the 4 countries. In Table 2, we present the distribution of the different users in each country and the number of valid Spanish tweets available. Valid tweets included Tweets in Spanish, as determined by Twitter's API, with more than 5 tokens after the pre-processing step. This included the removal of all trailing hashtags, as the weak-labels were assigned in part by leveraging the usage of labeled hashtags at the end of a tweet.

**Table 1. Example of the most prominent stance-tags used for each country. Pro-government hashtags tend to support the country's president (Evo, Piñera, Lenin), the police (ESMAD or Carabineros) or attack the opposition leader (Petro, Correa, CONAIE). Against-government hashtags label the country's president as a dictator, assassin or traitor, or support the protest ("paro").**

|  | Pro Government | | Against Government | |
|---|---|---|---|---|
| Bolivia | #AbajoElGolpe | #ElMundoConEvo | #AutoGolpeDeEstado | #EvoDictador |
|  | #EvoNoEstasSolo | #GolpeDeEstadoEnBolivia | #CarlosMesaPresidente | #BoliviaNoHayGolpe |
| Chile | #YoApoyoACarabineros | #FuerzaPiñera | #CarabinerosAsesinos | #FueraPiñera |
|  | #FueraComunismoDeChile | #DejenTrabajar | #AsambleaConstituyenteoNADA | #ChileEnDictadura |
| Colombia | #ApoyoAlESMAD | #ApoyoAlPresidente | #ESMADAsesino | #DuqueAsesino |
|  | #NoMasPetro | #ColombiaNoPara | #VivaElParoNacional | #ChaoDuque |
| Ecuador | #LeninNoCedas | #CONAIEterroristas | #DictaduraEnEcuador | #LeninChao |
|  | #FueraComunistas | #NoAlGolpeCorreista | #ElParoNoPara | #LeninMorenoTraidor |

**Table 2. Distribution of labeled users, their first-order neighbors (based on the response network) and their tweets (including retweets) for each of the countries studied and the 2020 Chilean Constitutional Referendum.**

| | Users | | | Tweets | | |
|---|---|---|---|---|---|---|
| | **Against** | **Pro** | **Neighbors** | **Against** | **Pro** | **Neighbors** |
| **Bolivia** | 58,508 | 54,347 | 292,684 | 3,508,300 | 3,583,943 | 12,507,155 |
| **Chile** | 220,391 | 33,331 | 409,014 | 14,659,535 | 2,775,496 | 14,211,964 |
| **Colombia** | 79,874 | 28,322 | 257,912 | 5,328,651 | 1,983,161 | 7,501,917 |
| **Ecuador** | 51,466 | 25,567 | 170,780 | 3,352,028 | 1,336,546 | 5,963,914 |
| **Chilean Referendum** | 10,423 | 11,206 | 96,202 | 6,454,647 | 6,060,679 | 1,288,351 |

For training purposes, we constructed training sets for each country by sampling 80% of the users (with their corresponding tweets), using 10% for validation and a 10% held-out test set. Splitting at the user level guarantees that there are no duplicate users between the different splits. Moreover, even though some retweets posted by users might target tweets in different splits, we ensured the corresponding validation and test nodes were masked during training. This is consistent with a transductive learning setting, which is common practice when training GNNs [29].

It is important to note that given the temporal proximity of the protests, and their corresponding collection dates, there is a non-trivial percentage of users that were active during more than one event. In Table 3, we present the percentage of users collected for each country (column) that were also contained in the training set of the other country (row). The largest overlaps occur when comparing the Chilean collection with the other countries, as users involved in this discussion represent more than 20% of the user base of other countries. This is expected considering that it involved at least twice as many users as the other collections. For all other country pairs, the overlap varies between 8 and 17 percent. However, this percentage diminishes considerably when considering tweets that formed part of multiple collections. As we see in Table 3, the overlap with the Chilean collection drops an average of 13 percentage points, while for all other country pairs their similarity is now below 7 percent (Ecuador and Colombia share no tweets which are consistent with their collection dates). To avoid bias in our results, given that one of the objectives of this study is to evaluate the extrapolation capabilities of models trained at different levels of user contextual user information, we mask users seen in multiple collections when performing cross-country experiments.

Although the regional nature of the dataset allows us to test the robustness of our models at different levels of contextual user information used, it does not allow us to effectively test this robustness over time. To address this issue, we constructed an additional weakly-labeled dataset around the 2020 Chilean Referendum.

**Table 3. Percentage of users, and their respective tweets, in the protest data of country *j* (column) that were also present in the training set of a country *i* (row).**

| | Bolivia | | Chile | | Colombia | | Ecuador | | Chilean Referendum | |
|---|---|---|---|---|---|---|---|---|---|---|
| | **Users** | **Tweets** | **Users** | **Tweets** | **Users** | **Tweets** | **Users** | **Tweets** | **Users** | **Tweets** |
| **Bolivia** | — | — | 11.45 | 6.33 | 13.54 | 4.47 | 23.99 | 5.67 | — | — |
| **Chile** | 25.79 | 18.74 | — | — | 20.2 | 8.26 | 30.05 | 13.36 | 43.90 | 0.00 |
| **Colombia** | 13.10 | 4.58 | 8.60 | 2.85 | — | — | 17.00 | 0.00 | — | — |
| **Ecuador** | 16.44 | 2.92 | 9.10 | 2.30 | 12.08 | 0.00 | — | — | — | — |

## 3.1 The 2020 Chilean Plebiscite

Throughout the 2019 Chilean protests, different social movements made calls in favor of drafting a new constitution. This social pressure reached a boiling point in November of that same year, which led the Chilean National Congress to agree to hold a National Plebiscite. The Plebiscite was scheduled for the start of 2020 but was delayed because of the Coronavirus pandemic. In October of 2020, it was overwhelmingly approved with 78% of the vote. Using Twitter's v2 full-archive search endpoint feature which was available on their Academic Research Track, we collected tweets from September 25 to November 10 of 2020 (a month prior and two weeks after the plebiscite took place). This research program allowed full historical access to publicly available tweets matching complex queries. The queries matched 124 hashtags and terms relevant to the event (e.g.: *#Plebiscito2020*, *#PlebiscitoChile*, *#NuevaConstitucion*, etc.). We then labeled these hashtags to identify useful "Stance Tags".

The labeling procedure was as follows: an expert in the South American region who is fluent in Spanish reviewed a sample of tweets at the end of the collection period. The annotator then established if the tweets were used consistently in favor or against the approval of the new Constitution (or of the referendum process in general), or if this was not possible the hashtag was labeled "Undetermined". This was done in separate meetings with the authors of this paper, where the reasoning for each label was openly discussed. This process resulted in 27 "Undetermined", 64 "Against", and 32 "Approve" stance tags. Instead of using hand-labeled Political Figures for the second stance signal, as was done for the construction of the protest data [7], we opted to identify the hashtags used in the user description that were explicitly rejecting or approving the referendum in first person. This resulted in 16 "Approve" (*#Apruebo*, *#AprueboCC*, etc.) and 21 that "Against" (*#YoRechazo*, *#Rechazo*, etc.). This set is used to label users based on their user descriptions. We found that the usage of stance tags in the descriptions as a replacement signal, despite requiring less annotation effort than labeling political figures, also resulted in meaningful polarized stance partitions based on the same language metrics used to validate the protest weak-labels.

Following the previously described methodology for the construction of the South American protest data [7], tweets or descriptions were assigned a stance if they used stance tags with consistent stances. We only proceeded with users that had at least 8 tweets with a consistent stance or if at least one description was consistent. A stance was assigned to a user if at least 90% of their tweets had the same stance. For description-based stances, a user was assigned a stance if all different user descriptions observed during the period were assigned the same stance. The final user stance was determined by combining the labels obtained from both sources and validating the stance assigned to a subset of the users. The final number of user counts and their distributions are presented in Table 2. Moreover, as shown in Table 3, 43.89% of users labeled were part of the 2019 Chilean protest training set while, by construction, no tweets were part of the protest data. We also note that only 0.015% of retweets that occurred in the referendum collection reference a tweet that occurred during the protests (this is consistent with the diffusion dynamics of retweets [30]). Moreover, to improve the resolution of the networks available for the labeled users, we collected their timelines during the event. Timelines were not always collected in the original protest data, but we hypothesize that the additional timeline context for each user will improve the quality of user embeddings for the referendum data.

Following X's January 2023 User Protection Policy update, tweet IDs related to sensitive political events cannot be publicly shared. We respect this policy, and instead share only anonymized network edges, the type of tweet the edge represents (Original, Reply, or Quote),

and user weights for replication. These resources can be found at: https://doi.org/10.5281/zenodo.14207926. The code for this paper is publicly released at https://github.com/rvillaco/Protest_Stance_Detection and the weights for the trained models are available at https://doi.org/10.5281/zenodo.15571840.

# 4 Materials and methods

The focus of this work is to explore the effect of increasing the level of contextual user information available to a classifier in the task of political target-stance classification. In line with this goal, we build a compartmentalized architecture that allows the evaluation of component outputs at various levels of contextual information leveraged.

In Fig 1, we present the different context-based components of the architecture developed. Note that the output of each component not only serves as input for the next, but can be used for stance prediction with the information available at its contextual level. Training and predictions for each component were done independently. In training, we optimize using cross-entropy loss using Adam with weight decay, a linear schedule with warmup and a maximum learning rate of 2e-4 (1e-3 for the network component). We train all models over 10 epochs with early stopping based on validation loss. Due to the unbalanced nature of the dataset in most countries (e.g.: Chile's label distribution was 87–13%), we opted for a dynamic resampling approach (with replacement) that weighted tweets and users based on the inverse frequency of their corresponding label. This under-samples the majority label and over-samples the minority label, and is performed at the start of each epoch. We found that this strategy improved training stability and the validation performance of our transformer models when compared to other static over and under-sampling strategies or when weighting the loss function by the inverse class frequency. We hypothesize that this is due to the inclusion of the

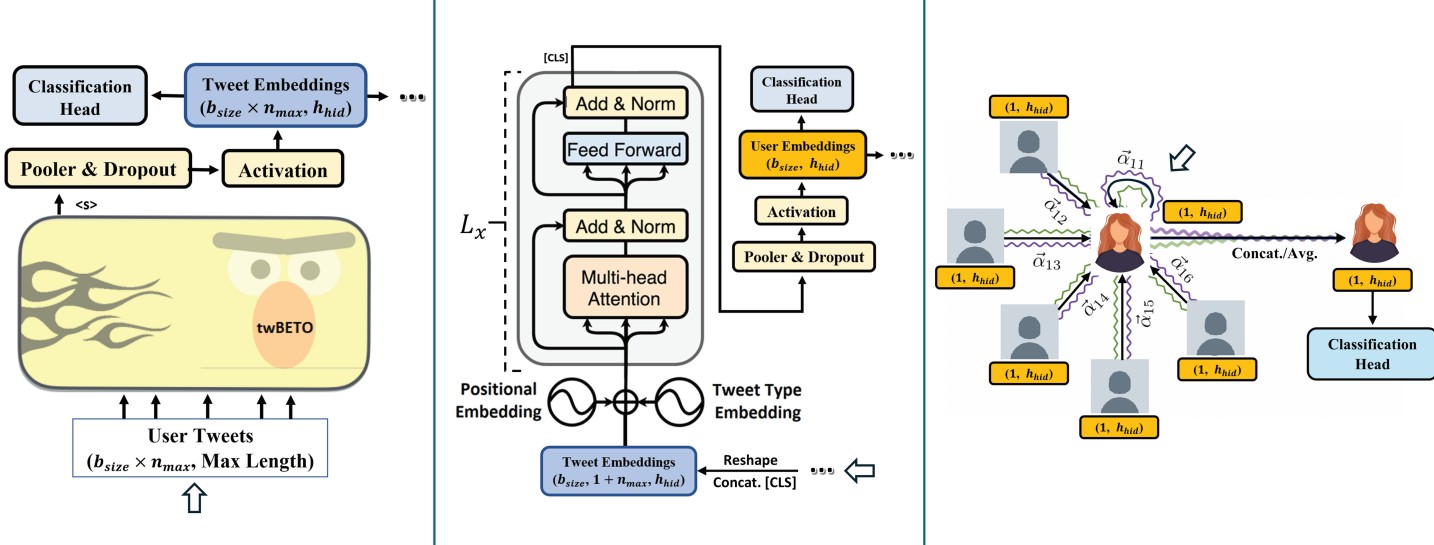

**Fig 1. Proposed compartmentalized architecture for Context-Sensitive Stance Classification.** The white arrow denotes the starting input for each component. The left figure presents the Tweet Level encoding applied to a batch of $n_{max}$ tweets from a given user producing a vector of dimensions $(n_{max}, h_{hid})$ per user. For the *Tweet Embeddings* to serve as input for the User Transformer (depicted in the middle panel) they are reshaped and a [*CLS*] parameter vector is appended at the start of each user's tweets. In this layer, a batch of $b_{size}$ users is processed by $L_x$ encoder stacks obtaining the final *User Embeddings*. On the right, we show the heterogeneous GAT model, wherein a user receives information from its neighborhood $\mathcal{N}(i)$ and their corresponding *User Embeddings*. The attention mechanism learns different attention weights $\alpha_{i,j}$ for $j \in \mathcal{N}(I)$ which reflect the importance of a neighbor $j$ to the label of the $i$'th user.

$n_{max}$ hyperparameter which limits the number of tweets considered for a user in a given batch (see Sect 4.2 for more details). In what follows, we describe the different components of our proposed architecture. As our focus is on exploring the effects of integrating contextual user information and not on training the best possible model for each country, we used only the more balanced Bolivian dataset to tune hyperparameters and select architectures.

## 4.1 Tweet encoder

The first component of the model creates embeddings for the different tweets produced by the users. For this purpose we pre-trained a BERT [8] language model, which we call *TwBETO v0*, following the robust approach introduced in RoBERTa [31]. We opted for the smaller architecture dimensions introduced in DistilBERT [32], namely, 6 hidden layers with 12 attention heads and a hidden dimension of 768. We also reduce the model's maximum sequence length to 128 tokens, following another BERT instantiation trained on English Twitter data [33]. We utilize the RoBERTa implementation in the Hugging Face library [34] and optimize the model using Adam with weight decay, a linear schedule with warmup and a maximum learning rate of 2e-4. We use a global batch size (via gradient accumulation) of 5k across 4 Titan XP GPUs (12 GB RAM each) and trained the model for 650 hours.

The model was trained with a corpus comprised of 155M Spanish tweets (4.5B words tokens), as determined by Twitter's API, and includes only original tweets (retweets are filtered out) with more than 6 tokens, while long tweets were truncated to 64-word tokens. The data was compiled from the following sources:

- 110M Tweets (3B word tokens) from the South American protests collected from September 20 to December 31 of 2019.
- 25M (0.7B word tokens) Tweets collected around the Coronavirus pandemic from April 01 to December 31 of 2020.
- 3M (0.3B word tokens) Tweets collected around the Chilean referendum from September 25 to November 10 of 2020.
- 17M (0.5B word tokens) rehydrated targets across all the collections listed.

Tweets are pretokenized using the "TweetTokenizer" from the NLTK toolkit [35] and use the emoji package to translate emotion icons into word tokens (in Spanish). We also preprocess the Tweets by replacing user mentions with "*USER_AT*" and, using the tweet JSON, we replace media urls with "*HTTPMEDIA*" and web urls with "*HTTPURL*". We found that this new model produced significantly better quality embeddings than the only other currently available Spanish BERT variant for Twitter [9]. We hypothesize that this is a result of the latter being trained mainly in European Spanish with fewer data and it not applying the RoBERTa pretraining framework. We make the pretrained *TwBETO v0* language model available through the Hugging Face model hub (https://huggingface.co/Ramavill/twBETO_v0).

The output of our *TwBETO* model, after being fed a batch of tweets, is pooled and passed through an activation layer. We use the standard pooling method used for BERT models, namely using the first element of the output of the final layer (corresponding to the "⟨s⟩" token), as input for a fully connected layer and passing through an activation. This output serves as our final *Tweet Embeddings* of size $h_{hid}$ = 768 per tweet. For the Tweet-level Context prediction, the embeddings are used as input for a classification head comprising a fully-connected layer and a softmax to predict the stance of a batch of tweets. Given that our main focus is to evaluate the effect of increasing the contextual user information available to the model, we do not fine-tune the *TwBETO* parameters in any training settings.

## 4.2 User encoder

As shown in the center part of Fig 1, the second component of our architecture is comprised of a stack of Transformer Encoder blocks [36] which operate on the *Tweet Embeddings* for a given user. As this requires a fixed input size, we introduced a parameter $n_{max}$ that determines the maximum number of tweets to consider for each user. In this work, we use $n_{max}$ = 15. When users exceeded this limit, we sampled $n_{max}$ tweets before assigning them to a batch. This way, we avoided wasting information as different tweets can be included each time a user is sampled. We hypothesize that this, combined with the resampling approach taken to account for class imbalance, improved training stability as repeated users had a high probability of including different tweets (at least 83% of the users in each training set had 15 or more tweets in the dataset). Given the power law distribution of user tweet counts and to reduce sampling times when constructing the batches, we chose only to keep the first 150 tweets available for each user. For the *Tweet Embeddings* to serve as input for this component we reshape them and append a [*CLS*] parameter vector at the start of each user's tweets, obtaining a tensor with dimensions: $b_{size}, 1 + n_{max}, h_{hid}$. Where $b_{size}$ is the number of users included in the batch and $n_{max} \leq N$ is the aforementioned maximum number of tweets allowed per user in the batch. We used trainable positional embeddings to maintain the temporality of the tweets and introduced a second type of trained embeddings to encode the type of each tweet as Original, Reply or Quote. Twitter users can interact in various ways and we hypothesized that different interaction types would have different impacts on stance. This was confirmed in our ablation study (done for Bolivia), as the inclusion of each of these components improved the validation macro-F1 score at this context level. These 3 embedding tensors are then added and normalized (by layer normalization) and serve as input for our encoder stack. Based on validation results, we use an encoder stack of size $L_x$ = 3, an intermediate embedding size of 2048, 6 attention heads and a GELU activation for the transformer stacks.

The output of the last encoder layer is pooled following the same strategy used for the *TwBETO* model as this allows the [*CLS*] parameter to attend to all tweets in the sequence and be optimized for stance classification. The output of this pooling layer serves as our final *User Embeddings* which maintain a dimension size of $h_{hid}$ = 768. As before we feed the embeddings to a classification head to perform user-level classification, or use this as input for the next component or our architecture.

## 4.3 Network-based prediction

We take the embeddings produced by the user encoder and predict users' stances using their interactions in the social network. We explored several different graph neural network architectures, including GraphSage layers [37], Graph Convolutional layers [38], and W-L Graph Convolutional layers [39]. Ultimately, we achieved the highest macro F1 score using Graph Attention Network (GAT) layers [40]. In the social setting imposed by Twitter's platform, there are also different user information context levels that can be leveraged by our Network-based classifier to improve stance detection. To test this, we trained separate homogeneous models for the network retweets, the network of responses (the union of replies and quotes) and the combined network (obtained by the union of both). Finally, we trained a heterogeneous model where we distinguish response and retweet edges.

For homogeneous graph experiments, after testing different alternatives, we use two GAT Layers with dropout, each with four attention heads, followed by 4 batch-normed linear layers with dropout and relu activations. The corresponding output is fed to a classification head for our final network-based user stance classification. We observed our heterogeneous model quickly overfit the data with our homogeneous architecture, so for the heterogeneous

model, we removed 2 linear layers and use only one attention head (for each edge type). Given the size of the underlying graphs, we sample neighborhoods for each node using the neighbor loader proposed in [37].

In our GAT models, for a given user $i$, the input to the attention layer is the *User Embeddings* of a random sample of $k$ one-hop neighbors of $i$ denoted as $\mathcal{N}_k(i)$. In our homogeneous GAT networks, the attention mechanism calculates the importance of $\mathcal{N}_k(i)$ to the label of $i$. In our heterogeneous GAT networks, the attention mechanism calculates the $\mathcal{N}_k(i)$ with relation $\Phi$ (retweet or response) to the label of $i$.

## 4.4 Baselines

To demonstrate the overall effectiveness of our approach, we sought to re-implement as baselines four recent user-level stance detection papers on our data that both leverage user-interaction networks. However, two of these approaches were not possible to re-implement as we discuss next. The first proposed an unsupervised clustering methodology for user-level stance detection [15], that did not scale to the dataset explored in this work as it requires pairwise similarity calculations for all users of interest. The second approach proposed a multi-modal cross-target stance detection architecture [24] that uses follower-ship and friendship networks for each user which were not possible to recover due to API constraints.

The first approach that we successfully adapted to our dataset was TSPA [18], which uses label propagation first over the retweet network, and then over the reply network. We implement the TSPA version without weighted propagation as it achieves comparable performance to weighted propagation, and it provides us with a baseline that uses only network information. This helps us understand the signals associated with the natural community structure that exists within the network. As the original authors, we were unable to run the algorithm on the Chilean dataset given its size. We followed the same pre-labeling approach used by the authors, but note that not all labeled users could be assigned predictions as some components of the interaction for each country did not receive labels. For fairness of comparison, we evaluate the accuracy and F1 only on users for which the algorithm assigned labels (a total of only 21 users across the Bolivia, Colombia and Ecuador test sets were unlabeled). As TSPA is an extension of the label-propagation algorithm, cross-country baselines are not possible with the algorithm. TSPA uses only network signals, and apart from initial hashtag label seeds, does not incorporate any external user text or features.

The final baseline implemented was Retweet-BERT [19], which performs unsupervised training on Sentence-BERT to contrastively embed user descriptions based on user retweet networks. They then fine-tune this trained model on user political stance task. As not all users in our data have descriptions, and many user descriptions changed, we implement the unsupervised fine-tuning step only on the first description for each labeled user using the retweet network of all labeled users. We then fine-tune this model on training users with descriptions and evaluate the model on validation and test users with descriptions. As users were randomly split into train, validation, and test sets, this approach impacted each split roughly equally. In each country between 70% and 80% of labeled users had descriptions. As data were weakly labeled using hashtags, hashtags could provide too easy a signal. Consequently, all hashtags and urls were removed from tweets and user descriptions baselines and for our proposed architectures.

## 5 Results & discussion

### 5.1 Main results

In this section, we present the results of the different components of our architecture to evaluate the effects of increasing the level of context available to a classifier. We use the Tweet-level classification as a baseline for our ablation studies. It is important to note that this task is evaluated at a tweet level (not at a user level) by assigning each its corresponding user stance. We define four different baseline experiments to assess the performance of classifiers trained at this context level:

- *Weak-Labeled Tweets*: This model was trained using tweets that contained "Stance Tags" as identified in [7]. All trailing hashtags were removed during pre-processing to avoid overfitting to these tokens.
- *All Original Tweets*: This model was trained using all original, replies and quote tweets produced by the different weakly-labeled users.
- *User Average of Original Tweets*: This model was trained by averaging the predicted labels assigned to a given user. A user is assigned the majority of their predicted labels. When no majority exists, a stance is assigned randomly.
- *User Average of All Tweets*: This User-Level prediction exercise is similar to the one described above, but also includes the retweets of a user. Retweets are assigned the label predicted for their target.

We include the last two user-level predictions to evaluate how effective our proposed User Transformer is at aggregating tweet information. We also include the TSPA model [18] (see Sect 4.4) as a network baseline that uses only a user's ego-network information. This helps us understand the signals associated with the natural community structure that exists within the network. However, as mentioned before, TSPA could not be run on Chile due to its data size. We evaluate the performance of each model using accuracy, macro F1 (due to the observed asymmetries in the label distribution), and the Area Under the Receiver Operating Characteristic (ROC AUC) on the held-out test set. The latter metric accounts for the possible differences in the optimal decision threshold that might exist across the different datasets. To assess whether the differences seen are statistically different, we recomputed the model performance metrics based on 1000 bootstrapped samples of the corresponding test set. We report the median result of the bootstrapped empirical distribution for each metric and compute its corresponding confidence band to assess whether the differences are statistically significant (these are presented in the Supporting Information).

In Table 4, we present the results of the ablation studies. As shown, there is a clear performance improvement when more contextual information is available to the classifier. As expected, focusing only on the weakly-labeled tweets presents an easier task than focusing on all original tweets, but still falls short of User-Level averages. When retweets are included, the performance improves but is still significantly below the User Transformer in all countries and across all metrics (see Table 1 of S1 for the 95% confidence bands of each metric). Interestingly, we observed that, for Colombia, adding retweets hurts the user-average performance. The User Transformer does not exhibit these disparate behaviors, highlighting its ability to identify tweets that are more relevant to the stance of the user. While the user-level transformer significantly outperformed Retweet-BERT [19] in all settings, it yielded statistically significant lower accuracy and M-F1 scores than TSPA in Bolivia and Ecuador. In the case of Colombia, the differences are not significantly different, which is consistent with what was noted before for the Retweet Average baseline. TSPA's relatively strong performance

**Table 4. Performance of in-country Stance Classifiers at different context levels.** Model performance metrics correspond to the median of 1000 bootstrapped samples of the test set. For each metric, the classifiers whose performance is not statistically different, at a 95% confidence level, from the best-performing model are highlighted in bold. We also highlight in bold the name of the best-performing model if it obtains the best results (or is statistically tied) in all presented metrics. The corresponding confidence bands, computed from the bootstrapped empirical distribution of each metric, are included in Table 1 of S1.

| | | | Chile (%) | | | Bolivia (%) | | | Ecuador (%) | | | Colombia (%) | | |
|---|---|---|---|---|---|---|---|---|---|---|---|---|---|---|
| | | | Acc. | M-F1 | ROC AUC | Acc. | M-F1 | ROC AUC | Acc. | M-F1 | ROC AUC | Acc. | M-F1 | ROC AUC |
| Tweet | Weak-Labeled Tweets | | 85.48 | 83.90 | 85.60 | 83.75 | 73.82 | 88.38 | 77.42 | 76.53 | 79.70 | 82.47 | 82.14 | 82.97 |
| Level | All Original Tweets | | 75.50 | 75.49 | 76.28 | 72.03 | 66.83 | 77.44 | 69.95 | 65.55 | 72.61 | 73.45 | 71.12 | 83.92 |
| User-Level | Avg. | All Original | 86.07 | 75.99 | 86.19 | 79.53 | 79.18 | 83.93 | 81.40 | 78.45 | 83.80 | 80.72 | 73.66 | 81.21 |
| | Avg. | With Retweets | 94.79 | 88.70 | 95.77 | 89.03 | 88.97 | 95.71 | 92.30 | 90.60 | 95.81 | 81.41 | 67.71 | 93.02 |
| | Retweet-BERT | | 92.84 | 83.42 | 92.63 | 75.77 | 75.77 | 84.11 | 71.74 | 63.03 | 73.70 | 79.25 | 72.43 | 80.01 |
| | Transformer | | 95.98 | 91.53 | 97.71 | 94.06 | 94.05 | **96.60** | 95.00 | 94.35 | **98.11** | **95.61** | 94.24 | 98.24 |
| Network-Level | Homogeneous | TSPA | — | — | — | **95.02** | **95.02** | 95.55 | **95.87** | **95.33** | 96.89 | **96.34** | 95.19 | 94.29 |
| | | Response | 95.61 | 90.58 | 97.69 | 94.01 | 93.94 | **96.82** | 94.20 | 92.89 | 97.65 | 95.11 | 93.57 | 98.22 |
| | | Retweet | 96.23 | 91.54 | 97.77 | 94.00 | 93.99 | **96.56** | 95.03 | 94.39 | **98.13** | 95.68 | 94.39 | **98.44** |
| | | Combined Network | 95.74 | 90.00 | 97.70 | 94.06 | 94.06 | **96.58** | 94.66 | 94.02 | **98.13** | 95.60 | 94.23 | **98.46** |
| | Heterogeneous | | **97.41** | **94.41** | **98.62** | **95.76** | **95.76** | **97.01** | **96.22** | **95.76** | **98.64** | **97.01** | **96.08** | **98.91** |

demonstrates the strong stance signals present in each network, even without user text features. Nonetheless, the User Transformer significantly outperforms it, across all countries, when considering their ROC AUC score, which shows that no model uniformly outcompetes their counterpart. Importantly, and as hypothesized at the start of this work, leveraging the ego-network of the user, in conjunction with the semantic features aggregated by the User-level Transformer, can consistently improve upon its already high bar. In all four countries, the heterogeneous model, which leverages all network information, yielded macro F1 scores greater than 94.1 and accuracy scores greater than 95.7. The difference in model performance is significantly different in Chile while for the remaining countries, other network models or the User Transformer can match its performance in some metric. However, as we will explore next, the differences in performance are exacerbated when exploring each model's generalization capabilities.

## 5.2 Robustness analysis

**Cross-country robustness.** Even though the addition of social context can lead to an in-sample increase in classification performance, we also seek to evaluate how context can affect the generalizability of our proposed classifiers. Given that our target countries share a common language, we hypothesized that the models should be able to extrapolate the stance learned from one country to another. Protests in Bolivia had opposing ideological motivations in comparison with the motivations observed in the other countries, which we hypothesized would degrade cross-country performance significantly. The results for the cross-country ablation are shown in Table 5. For these experiments, we excluded from the test country (column) any user that was part of the training set of the other country's classifier (row). Refer to Table 3 for more details on the overlap between the different collections. As before, we recomputed the model performance metrics based on 1000 bootstrapped samples of the corresponding cross-country test set. For the economy of space, we only include the macro F1 score and the ROC AUC metric for the best-performing classifiers at each context level. As mentioned in Sect 4.4, we do not include the TSPA baseline as it is an extension of a label propagation algorithm that relies on hashtag seeds that are not reliably used in different protest collections.

As shown, the Bolivian case serves as an adversarial setting for classifiers trained in other countries, which suggests that, when applied to this country, the semantic features leveraged by the classifiers are operating on an ideological dimension. This is consistent with results presented in [7], where the authors found that language polarization remained along ideological lines when comparing protests of any of the three countries with Bolivia. This was not the case when comparing protests of the other three countries. As expected, Chile, Colombia, and Ecuador exhibit strong pairwise performance consistent with their ideological alignment. We can also note that the Tweet-Level performance varies significantly, falling in some cases 30 points below the user Transformer's macro F1, which is far more stable (73–85% in these countries). However, when contrasting their ROC AUC performance, both models remain competitive, which suggests that there are country-specific thresholds that can significantly improve the average tweet-level classification performance. Nonetheless, given that we seek to evaluate the one-shot performance of the different models (including its optimal decision threshold for the respective country), we will focus mainly on differences in macro F1 performance for the robustness analysis.

We found that on average, excluding Bolivian predictions, the heterogeneous network model increased cross-country prediction macro F1 score by 7.62 points and significantly outperforms all other models in all metrics and all country combinations except for one

**Table 5. Macro F-1 (%) score and Area Under de ROC curve for out-of-sample cross-country predictions for classifiers at different context levels. Model performance metrics correspond to the median of 1000 bootstrapped samples from column-country. This excluded users seen during training of the country classifier (row). For each metric, the classifiers whose performance is not statistically different, at a 99% confidence level, from the best-performing model are highlighted in bold. We also highlight in bold the name of the best-performing model if it obtains the best results (or is statistically tied) in all presented metrics. For the Bolivian case study, the worst-performing classifier is highlighted. The corresponding confidence bands, computed from the bootstrapped empirical distribution of each metric, are included in Table 2 of S1.**

| | | Chile | | Bolivia | | Ecuador | | Colombia | |
|---|---|---|---|---|---|---|---|---|---|
| | | M-F1 | ROC AUC | M-F1 | ROC AUC | M-F1 | ROC AUC | M-F1 | ROC AUC |
| **Chile** | Avg. Tweet-Level (WR) | — | — | 37.37 | 19.68 | 57.67 | 88.05 | 89.92 | 95.98 |
| | User-Level Transformer | — | — | 35.73 | 27.29 | 81.24 | 88.66 | 84.32 | 93.31 |
| | **Hetero. Network-Level** | — | — | **22.81** | **18.36** | **88.95** | **97.23** | **92.57** | **98.49** |
| **Bolivia** | Avg. Tweet-Level (WR) | 9.86 | 11.78 | — | — | 26.84 | 27.69 | 17.36 | 14.78 |
| | User-Level Transformer | 9.94 | 18.74 | — | — | **25.26** | 45.41 | 18.36 | 24.06 |
| | **Hetero. Network-Level** | **3.12** | **7.53** | — | — | **25.74** | **24.89** | **5.41** | **9.52** |
| **Ecuador** | Avg. Tweet-Level (WR) | 58.18 | 93.63 | 33.87 | 13.13 | — | — | 80.01 | 86.44 |
| | User-Level Transformer | 80.43 | 90.87 | 23.57 | 15.98 | — | — | 77.90 | 89.75 |
| | **Hetero. Network-Level** | **92.13** | **97.17** | **9.80** | **5.08** | — | — | **95.50** | **98.33** |
| **Colombia** | Avg. Tweet-Level (WR) | 80.86 | 91.93 | **39.05** | 38.79 | 45.69 | 80.26 | — | — |
| | User-Level Transformer | 73.48 | 93.26 | **38.59** | 29.92 | **77.61** | 86.30 | — | — |
| | **Hetero. Network-Level** | **85.59** | **96.51** | 42.37 | **27.49** | 73.20 | **92.90** | — | — |

discussed below. In the best case, the Ecuadorian instance yielded a zero-shot macro F1 of 95.5 on Colombian data, improving on its User Transformer counterpart by 15 points. Interestingly enough, the only case where the heterogeneous model underperforms its User Transformer counterpart (by 4 points) is when applying the Colombian models to Ecuadorian data. It is worth noting that although the heterogeneous model achieves this in countries with similarly motivated protests, it also significantly underperforms other alternatives in the Bolivian case (except for the Colombian classifiers). This is consistent with the observed tendency of deep learning models to be overconfident in their predictions and highlights the importance of domain knowledge when applying these models to different domains. If we leverage the knowledge of the opposed motivations for the Bolivian protests by inverting the heterogeneous model's predictions, its performance would be in line with what was observed for the other countries. We take the approach of inverting the predictions when we report the results of the 2020 Chilean referendum.

**Robustness over time.** To test the effect of context over time, we applied the Chilean protests classifiers to predict the stance of users, in a zero-shot setting, towards the 2020 Plebiscite vote for drafting a new constitution. In contrast to previous experiments, we report the results for all users, regardless of whether or not they were seen when training the Chilean models. However, we segment the results based on this condition, as it allows us to contrast the performance of the different models on new users. We take this approach given that (1) it is often realistic on applied settings and (2) there is no overlap between the tweets seen in both collections (see Table 3). Additionally, only 0.015% of retweets in the referendum collection reference a tweet that occurred during the protests.

We hypothesized that opposition to the government during the protests should signal endorsement for the new constitution (the models should be good inverse classifiers). Table 6 presents the results of this exercise. We also include the predicted Referendum vote obtained when applying each classifier, based on the two-hop neighborhood of labeled users. The benefits of the heterogeneous model are more starkly observed in this setting, as it significantly outperforms all other variants across all metrics. However, surprisingly, the homogeneous network models underperform the User-level Transformer in all settings and these differences are statistically significant at a 99% confidence level. We hypothesize that this might be due

**Table 6. Out of sample Predictions for the Chilean Referendum at different context levels.** These correspond to the classifier trained on the 2019 Chilean Protest Data, but with inverted labels ("Pro" government are considered "Against" the referendum and vice-versa). Model performance metrics correspond to the median of 1000 bootstrapped samples of the Referendum collection, disaggregated by whether a user was seen during training of the Chilean Protest classifier. For each metric, the classifiers whose performance is not statistically different, at a 99% confidence level, from the best-performing model are highlighted in bold. We also highlight in bold the name of the best-performing model if it obtains the best results (or is statistically tied) in all presented metrics. The corresponding confidence bands, computed from the bootstrapped empirical distribution of each metric, are included in Table 3 of S1.

| | | | Accuracy (%) | | M-F1 (%) | | ROC AUC | | Pr-Ref (%) |
|---|---|---|---|---|---|---|---|---|---|
| | | | New Users | Protests User | New Users | Protests User | New Users | Protests User | |
| **Tweet-Level** | Weak-Labeled Tweets | | 69.74 | 70.70 | 61.70 | 70.13 | 78.97 | 79.09 | 57.82 |
| | All Original Tweets | | 64.48 | 70.75 | 63.68 | 67.69 | 72.61 | 74.61 | 65.18 |
| **User-Level** | Avg. Tweet-Level | All Original | 77.50 | 81.61 | 77.25 | 79.74 | 82.90 | 84.17 | 63.81 |
| | | With Retweets | 80.83 | 88.59 | 80.83 | 86.00 | 97.14 | 97.90 | 67.81 |
| | Transformer | | 89.43 | 91.19 | 89.18 | 90.24 | 95.97 | 96.56 | 63.36 |
| **Network-Level** | Homogeneous | Response | 76.04 | 84.78 | 76.03 | 81.26 | 93.34 | 93.65 | 86.3 |
| | | Retweet | 80.12 | 87.64 | 80.12 | 85.11 | 95.51 | 96.58 | 82.3 |
| | | Combined Network | 71.84 | 82.19 | 71.69 | 77.45 | 90.98 | 91.65 | 87.6 |
| | **Heterogeneous** | | **98.78** | **99.43** | **98.74** | **99.36** | **99.84** | **99.85** | **78.5** |

in part to differences in network construction: the Referendum data included the timeline of all users collected, whereas this was not done for the country-level protest data. This implies that the networks constructed for the former dataset are denser and include more reply and quote interactions between users. Hence, it more likely that we capture both positive and negative responses from one user to others of the same or different stances. While the User-level transformer can leverage the semantic context of each tweet as it pertains to the stance of the user, the homogenous models are forced to collapse these disparate signals into one interaction link. As such, we see that the Retweet-based homogenous model shows the least degradation (as retweets provide more consistent endorsements), while the combined network model drops significantly below other simple average user-level models. However, when the model is allowed to leverage the heterogeneous signals provided by retweets and other responses, its performance improves dramatically. Moreover, except for the heterogeneous network model, the performance of all classifiers is significantly reduced when dealing with completely new users at a 99% confidence level. This again highlights how the disparate signals present in a user's ego-network can improve the robustness of the classifier over time. The User-level Transformer shows the second least degradation, while other models show a 10-point drop in performance. With regards to the predicted referendum vote (shown as "Pr-Ref" in Table 6), we note that semantic-based classifiers tend to undershoot the observed Referendum tallies (the final vote was 78% in favor of the new constitution), while the single-network-based classifiers do the opposite. Interestingly, the heterogeneous model predicts the voting ratio almost perfectly which is encouraging. Albeit, we stress that given that there is no evidence that the sample constructed is representative of the Chilean population, the closeness of this prediction is merely suggestive of good performance and more research is necessary to assess if this behavior is truly generalizable. The network models' strong performance, when allowed to leverage the heterogeneous interactions between users, can be in part explained by the fact that language and conversation topics change faster than social ties, and the fact that social ties themselves can alter language [41]. As a result, features extracted from interaction networks may carry strong signals that are more generalizable to different cultural contexts.

To summarize, in all but one of the robustness experiments conducted, our heterogeneous graph neural network yielded the highest accuracy and F1 scores. The strong performance

of the heterogeneous model aligns with intuition, as different edges on Twitter have different social functions. Retweeting with no commentary is more likely an endorsement than an argument, whereas replying could imply a fight or it could imply an agreement. To further test this explanation, for each node $i$ in labeled Bolivian retweet and response networks, we calculated the percent of neighbors of node $i$ that shared the same label as node $i$ and then took the average of those percentages. In the retweet network, the average neighborhood-agreement percentage was 93%. In the Response (reply, quote, mention) networks, neighborhood agreement was only 69%. This suggests that ability to differentiate between relation types should help the model's performance, which was validated by the results of the referendum experiment. Our results confirm that including interaction types in models can yield modest improvements in predictive power in in-country stance detection tasks. However, we find that the inclusion of a user's heterogeneous ego-network information yields much larger improvements in related country-context assessments or its robustness over time.

## 6 Limitations and future work

There are several limitations to our study. First, we use binary stance labels in our models, so users with no or contradictory/nuanced opinions may be incorrectly categorized into pro- or anti-categories. Also, we did not attempt to remove bots or trolls, nor did we survey in-country Twitter user demographics, so these datasets may be noisy proxies for population opinions. However, despite this, we found the stance distribution of users in the Chilean referendum data was nearly identical to the final referendum vote. Nonetheless, given that the construction of the referendum data is in no way representative of the Chilean population at the time, further research needs to be conducted to assess whether it is truly generalizable. A possible avenue to explore would be to reweight the sample with estimated demographic information (see [42] for a promising approach to this problem). Finally, our network model only uses interactions and the text of posts for classification. This excludes many other attributes available on individual posts that might correlate with stance, including URLs, followership, posting patterns, bios, and shared multimedia. In future work, we plan to explore the integration of multimodal data as well as the integration of other attributes that may be useful in detecting stance.

## 7 Ethical considerations

We comply with Twitter's fair use policy by presenting our results at an aggregate level. We make no effort to identify individual users, and the primary interest of this work is to assess how classifier performance is affected when we leverage the different levels of contextual information available as users interact through social media. However, in this research, we demonstrate how indirect interactions on these platforms can still provide strong signals for identifying user stances, even in cases where users may not explicitly express their opinions. In other words, advertisers and governments may be able to target users using indirect ties.

In a best-case scenario, this might allow organizations to send users relevant advertisements or to provide beneficial information on government services to relevant groups. However, in worst-case scenarios, systems like this also have the potential to be abused, as corporations or governments could choose to target vulnerable or dissident groups. This further underscores the importance of social network platforms protecting user anonymity and privacy and the importance of being cognizant of traces one leaves online. We hope that our work can help inform future discussions on protecting user privacy.

## 8 Conclusions

In this work, we explored the value of contextual user information in the task of target-stance classification during the 2019 South American Protests. For this purpose, we constructed a compartmentalized architecture that relied on Transformers for the Tweet and User level contexts, and GNNs to leverage social media relations. We found that increasing context not only improved the performance of a classifier within the country it was trained but also made it more robust to out-of-sample predictions. We found these out-of-sample improvements were substantial both when comparing a classifier's performance across varying country contexts and over time.

## Acknowledgements

The collection of the data required for this work was possible due to access to Twitter's v2 full-archive search endpoint granted to us by their now discontinued Academic Research Program. We would also like to thank the two anonymous journal reviewers for helpful comments on an earlier draft of this work.

## Supporting information

**S1 Confidence Bands.** Confidence bands for the Macro F-1 score and Area Under the ROC curve of the within-country and out-of-sample cross-country predictions at different context levels. The confidence bands are computed from the corresponding percentiles of the 1000 bootstrapped samples constructed for each experiment.
(PDF)

## Author contributions

**Conceptualization:** Ramon Villa-Cox, Kathleen M. Carley.

**Data curation:** Ramon Villa-Cox, Evan M. Williams.

**Formal analysis:** Ramon Villa-Cox, Evan M. Williams.

**Funding acquisition:** Ramon Villa-Cox, Kathleen M. Carley.

**Investigation:** Ramon Villa-Cox, Evan M. Williams.

**Methodology:** Ramon Villa-Cox, Evan M. Williams.

**Project administration:** Ramon Villa-Cox.

**Resources:** Ramon Villa-Cox.

**Software:** Ramon Villa-Cox, Evan M. Williams.

**Supervision:** Kathleen M. Carley.

**Validation:** Ramon Villa-Cox, Evan M. Williams.

**Visualization:** Ramon Villa-Cox, Evan M. Williams.

**Writing – original draft:** Ramon Villa-Cox, Evan M. Williams.

**Writing – review & editing:** Ramon Villa-Cox, Evan M. Williams.

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
