## [Decision Letter · Decision Letter 0]

4 Oct 2024

PONE-D-24-21648Evaluating the Effect of Heterogeneous User Interactions in Out-of-Sample Stance ClassificationPLOS ONE

Dear Dr. Villa-Cox,

Thank you for submitting your manuscript to PLOS ONE. After careful consideration, we feel that it has merit but does not fully meet PLOS ONE’s publication criteria as it currently stands. Therefore, we invite you to submit a revised version of the manuscript that addresses the points raised during the review process.

We look forward to receiving your revised manuscript.

Kind regards,

Carlos Henrique Gomes Ferreira, Ph.D.

Academic Editor

PLOS ONE

Journal Requirements:

1. When submitting your revision, we need you to address these additional requirements.Please ensure that your manuscript meets PLOS ONE's style requirements, including those for file naming. The PLOS ONE style templates can be found at https://journals.plos.org/plosone/s/file?id=wjVg/PLOSOne_formatting_sample_main_body.pdf and https://journals.plos.org/plosone/s/file?id=ba62/PLOSOne_formatting_sample_title_authors_affiliations.pdf 2. We note that the grant information you provided in the ‘Funding Information’ and ‘Financial Disclosure’ sections do not match. When you resubmit, please ensure that you provide the correct grant numbers for the awards you received for your study in the ‘Funding Information’ section.

3. Thank you for stating the following in the Acknowledgments Section of your manuscript: "The research for this paper was supported in part by the ARMY Scalable Technologies for Social Cybersecurity, Office of Naval Research, MURI: Persuasion, Identity, & Morality in Social-Cyber Environments, and Office of Naval Research, MURI: Near Real Time Assessment of Emergent Complex Systems of Confederates under grants W911NF20D0002, N000142112749, and N000141712675. It was also supported by the Secretar´ıa de Educaci´on Superior, Ciencia, Tecnolog´ıa e Innovaci´on (SENESCYT), Ecuador; the Center for Informed Democracy and Social-cybersecurity (IDeaS) and the Center for Computational Analysis of Social and Organizational Systems (CASOS) at Carnegie Mellon University. The views and conclusions are those of the authors and should not be interpreted as representing the official policies, either expressed or implied, of the ARMY, the ONR, or the US Government".

Please remove any funding-related text from the manuscript and let us know how you would like to update your Funding Statement. Currently, your Funding Statement reads as follows: "The research for this paper was supported in part by the ARMY Scalable Technologies for Social Cybersecurity, Office of Naval Research, MURI: Persuasion, Identity, & Morality in Social-Cyber Environments, and Office of Naval Research, MURI: Near Real Time Assessment of Emergent Complex Systems of Confederates under grants W911NF20D0002, N000142112749, and N000141712675. RVC was also supported by the Secretaría de Educación Superior, Ciencia, Tecnología e Innovación (https://siau.senescyt.gob.ec/convocatorias), Ecuador; the Center for Informed Democracy and Social-cybersecurity (https://www.cmu.edu/ideas-social-cybersecurity) and the Center for Computational Analysis of Social and Organizational Systems (CASOS) at Carnegie Mellon University. The views and conclusions are those of the authors and should not be interpreted as representing the official policies, either expressed or implied, of the ARMY, the ONR, or the US or Ecuadorian Government. None of the sponsors or funders of this work played any role in the study design, data collection and analysis, decision to publish, or preparation of the manuscript.

Reviewers' comments:

Reviewer's Responses to Questions

**Comments to the Author**

1. Is the manuscript technically sound, and do the data support the conclusions?

Reviewer #1: Yes

Reviewer #2: Partly

2. Has the statistical analysis been performed appropriately and rigorously? 

Reviewer #1: No

Reviewer #2: Yes

3. Have the authors made all data underlying the findings in their manuscript fully available?

Reviewer #1: Yes

Reviewer #2: Yes

4. Is the manuscript presented in an intelligible fashion and written in standard English?

Reviewer #1: Yes

Reviewer #2: Yes

5. Review Comments to the Author

Reviewer #1: The manuscript investigates the impact of different levels of context in detecting the stance of social media users towards political events, with a specific focus on the 2019 South American Protests. It also evaluates the models' ability to extrapolate the prediction to the contexts of other countries. Authors have collected a large dataset of tweets in 2019 and an additional dataset in 2020 related to the Chilean Plebiscite. Different levels of context correspond to: tweet-level, user-level (i.e., including previous tweets), egonet-level (i.e., including interactions with other users). They pre-train a RoBERTA model to represent tweet content, a transformer-like model to represent user history and a heterogeneous Graph Attention Network to represent social media interactions and their types. The proposed model is compared with ReTweet-BERT and TSPA (a label propagation based algorithm with pre-calculated interaction weights). The proposed model outperforms the baselines and the additional levels of context have a positive impact on its performance when retweets are considered at the user-level and a heterogeneous GAT is used.

Strengths

S1. Large, exclusive dataset on the 2019 South American Protests and on the 2020 Chilean Plebiscite.

S2. A Spanish variant of BERT (twBETO) will be made available, trained on 150 million Spanish tweets

S3. Proposed model achieves excellent performance on political stance detection on the aforementioned dataset.

S4. Manuscript is well-written and easy to follow.

Weaknesses

W1. Title is more general than the work itself: the work is specific to political stance detection; other issues with title (see C1).

W2. Validation based on how closely they approximate the final vote of the Chilean Plebiscite has no statistical validity as they do not account for how representative the sample is (see C2).

W3. The cross-target comparison is based on Macro-F1, despite the different class proportions (see C3).

W4. A strong baseline of comparison was dismissed without clear evidence that it does not scale (see C4).

W5. Comparison between methods lacks confidence intervals (consider bootstrapping the test set).

W6. Lacks comparison with LLMs, e.g., LLaMa, GPT3.5, GPT4.0 (it would be good to include a comparison at least in the zero shot setting, but it is not essential for acceptance in my opinion).

Comments

C1. Current title is "Evaluating the Effect of Heterogeneous User Interactions in Out-of-Sample Stance Classification".

C1a. The use of "Stance Classification" is too general, and it may give the impression that the model can be applied to any stance detection task. Examples where it doesn't apply: detect if the author agrees or not with a previous comment; query if the author agrees with a given statement "X".

 Recommendation: use "Political Stance Detection".

C1b. Unclear what is meant by "effect of heterogeneous user interactions". The effect being evaluated is the **impact** on Political Stance Detection Performance. What is being evaluated is the **impact** of **accounting for**. It is hard to understand the meaning of 'heterogeneous' directly from the title.

C2. Given that there is no evidence that the user sample is representative of Chile's population, the fact that it is closer to the final vote cannot be used to claim that the proposed model is good. The correct way of doing this is by performing demographic inference and reweighing samples accordingly (see ref below). Authors could attempt to do this based on the author names and description with the due ethical considerations.

Zijian Wang, Scott Hale, David Ifeoluwa Adelani, Przemyslaw Grabowicz, Timo Hartman, Fabian Flöck, and David Jurgens. 2019. Demographic Inference and Representative Population Estimates from Multilingual Social Media Data. In The World Wide Web Conference (WWW '19). Association for Computing Machinery, New York, NY, USA, 2056–2067. https://doi.org/10.1145/3308558.3313684

Recommendation: either follow the work above to apply the correct weighting OR downplay the fact that the proposal model is close to the final vote and add this disclaimer.

C3. F1 scores are a function of the decision threshold. The optimal decision threshold varies based on the predicted scores distribution, hence it changes across datasets.

Recommendation: Authors should opt for metrics that are robust to the differences between domains, such as AUC or area under the precision-recall curve (AUPR).

C4. The manuscript states in l. 94 that:

"Even though there exists work that has explored user-level stance detection on Twitter data [15], the proposed approach requires pairwise similarity calculations for all users of interest, which limits the scalability of the approach"

Looking at Darwish et al. 2020, it seems that their algorithm depends on UMAP for dimensionality reduction and DBSCAN for clustering. Both algorithms have hyperparameters that allow them to scale by considering a smaller number of neighbors.

It is not clear that the algorithm cannot run on the present data. In fact, Darwish et al. have applied it to datasets of 1.8M, 2.4M and two datasets of 2.6M tweets, which are in the same scale as the datasets in the manuscript.

C5. Abstract refers to "users' social networks" which is a bit ambiguous. To distinguish from the platforms, authors should consider using the term 'egonets'.

C6. To deal with class imbalance, authors use resampling, which tends to leave out some examples from the majority class. Have you consider using class weights in the loss function?

Minor comments

l. 474 Undefined reference to Table 2.

Reviewer #2: In this paper, the authors propose a framework for stance detection based on three encoders (i.e., components) for content, user, and social network in the context of political protests, specifically the 2019 South America Protests.

Strength: The authors deal with a relevant and state-of-the-art research issue: evaluating layered stance detection approaches that go beyond text content. In this case, the text content (tweets), the user text pattern, and the user social network ties are explored. Additionally, the authors evaluate the impact of location and time on their framework performance by exploring data extracted from different countries.

Weakness: Some issues regarding the impact of each model component and the methodology to train and evaluate them must be clarified. First, the social network component and its emerging communities may alone answer the problem of user stance in favor or against protests. Second, the labeling process for training the model is superficially explained to make a self-contained manuscript.

In the following, I discuss the paper per section, including more details about the two issues reported above.

1. Introduction

Regarding the first contribution, I would recommend that the authors also mention the impact of each compartment of the proposed architecture on the stance-detection problem. As reported in the following comments, it is important to justify the construction of an architecture based on three compartments (tweet, user, and network layers), given its high computational cost.

2. Related Work

I recommend explaining specific technical terms such as “hand-labeled” and “hand-crafted” to make the manuscript clear for readers from broad research areas.

3. Data Description

Authors must provide further information about the adopted methodology to construct a ground truth dataset of labeled comments and evaluate the proposed framework. First of all, the weak-labeling step of the methodology must be explained in order to produce a self-contained manuscript. Questions that are raised from the current data description are:

- How is the “endorsement of hand-labeled political figures” leveraged?

- Which hashtag campaigns have well-defined stances toward each government? I recommend providing a table with prominent examples.

- Which polarized communities are observed according to user partitions from the constructed labels? It appears a network of users supports the weak-labeling methodology, but no information is available on how this network and connections between users are constructed.

The observation of polarized communities raises the issue of whether the users’ network and its emerging communities alone answer the problem of user stance in favor or against protests. What is the relevance (i.e., the impact) of the users’ text content for detecting their instances?

This issue should be addressed in the analyses, and the impact of each component on the stance problem in the context should be clarified.

Why the weak-labelling is not used for the Chilean Referendum dataset?

4. Materials and Methods

Some notes and recommendations for better comprehension of Fig1:

- Include input arrows for each layer to indicate where to start reading each layer, mainly the first and second ones, which have more details.

- Should the output dimension of the first layer (tweet encoder) be equal to the input dimension of the second layer (user encoder)? Please check it and explain it in the text.

- The Lx label's current position in the figure is difficult to understand. It must be checked.

- Labels in the third layer (network) must be explained.

- The quality of Figure 1, shown on page 26 of the reviewer file, is very bad. It is important to fix it.

Section 4.1: The dimensions of the user tweets input and the tweet embeddings output are not explained in the text. Please explain it.

5. Results

Regarding the results shown in Table 3, are the performances of the proposed User-Level Transformer and TSPA approach significantly different? A confidence interval for the performance metrics could be calculated to analyze whether they are significantly different. Approaches such as Normal Approximation Interval or Bootstrapping the Test Sets are currently good practices.

The network-level appears to improve the results significantly. Again, communities that emerge from the network, which were used to label users, are probably the most relevant information for user stance in favor or against protests. Thus, the result in Table 3 is the expected one (i.e., no novelties). Authors should discuss, based on their data, the relevance of exploring text content once you have the users-network, which is probably the most relevant information for stance in the analyzed context.

The results shown in Section 5.2 are very interesting. Specifically, the case of Bolivia shows that models trained with data from a given location cannot always be applied to other locations, e.g., countries, even though they share similar cultures and languages. On the other hand, the issue discussed above becomes more evident as cross-country results show that the users' network is the information that allows models to reach the best results (e.g., accuracy and F1 greater than 90% in most evaluated cases). Again, it is important to discuss the relevance of exploring text content once you have the users' network, considering the computing cost to deal with each one.

To correct the broken reference for Table 5 on page 13/18, line 474.

6. PLOS authors have the option to publish the peer review history of their article (what does this mean?). If published, this will include your full peer review and any attached files.

Reviewer #1: **Yes: **Fabricio Murai

Reviewer #2: No

---

## [Author Response · Author response to Decision Letter 1]

25 Nov 2024

Dear Reviewers and Editor,

We thank you for your thorough and helpful comments. We have attempted to address each; the comments have greatly improved the quality of the paper. Below we outline how we attempted to address comment point-by-point:

EDITOR COMMENTS

Corrected the title so it is in compliance with the style requirements.

Ensured the funding information in the Financial Disclosure sections is correct

Please remove any funding-related text from the manuscript and let us know how you would like to update your Funding Statement.

We removed all funding information from the 'Acknowledgments' section and updated it accordingly.

4. When completing the data availability statement of the submission form, you indicated that you will make your data available on acceptance. We strongly recommend all authors decide on a data sharing plan before acceptance, as the process can be lengthy and hold up publication timelines.

We comply with the suggestion and make all data publicly available before acceptance. The data and code can be accessed through the following URLs:

https://doi.org/10.5281/zenodo.14207926

https://huggingface.co/Ramavill/twBETO_v0

https://github.com/rvillaco/Protest_Stance_Detection

We removed all supporting information files from the submission as they are now freely accessible through the URLs mentioned above.

REVIEWER 1 RESPONSES

C1. Current title is "Evaluating the Effect of Heterogeneous User Interactions in Out-of-Sample Stance Classification".

C1a. The use of "Stance Classification" is too general, and it may give the impression that the model can be applied to any stance detection task. Examples where it doesn't apply: detect if the author agrees or not with a previous comment; query if the author agrees with a given statement "X"  Recommendation: use "Political Stance Detection".

C1b. Unclear what is meant by "effect of heterogeneous user interactions". The effect being evaluated is the **impact** on Political Stance Detection Performance. What is being evaluated is the **impact** of **accounting for**. It is hard to understand the meaning of 'heterogeneous' directly from the title.

We agree that the title was overly-general and appreciate the comments. We updated the title to “Social context in political stance detection: Impact and extrapolation”

C2. Given that there is no evidence that the user sample is representative of Chile's population, the fact that it is closer to the final vote cannot be used to claim that the proposed model is good. The correct way of doing this is by performing demographic inference and reweighing samples accordingly (see ref below). Authors could attempt to do this based on the author names and description with the due ethical considerations. Zijian Wang, Scott Hale, David Ifeoluwa Adelani, Przemyslaw Grabowicz, Timo Hartman, Fabian Flöck, and David Jurgens. 2019. Demographic Inference and Representative Population Estimates from Multilingual Social Media Data. In The World Wide Web Conference (WWW '19). Association for Computing Machinery, New York, NY, USA, 2056–2067. https://doi.org/10.1145/3308558.3313684 Recommendation: either follow the work above to apply the correct weighting OR downplay the fact that the proposal model is close to the final vote and add this disclaimer.

We agree with the observation and attempted to reweight the observations according to estimated demographic information. However, given the size of the referendum collection, the profile pictures were not saved when the collection took place. We attempted to download the pictures based on the saved JSONs but a large percentage of the user profile links were no longer available. Moreover, given the number of users (over 100k) and the new API rate limits imposed after the change from Twitter to X, it was not viable to attempt to download the profile pictures of the accounts that are still active (it is likely a considerable number of accounts no longer exist).

We tried to estimate the demographic information based on only on text features (replacing the pictures with random noise as provided in the paper’s repository) but the performance of the prediction dropped considerably below the ablation results provided in the paper (gender macro F1 dropped by 20%). We measured this based on a small sample of 50 accounts of politicians, reporters and institutions we constructed to test the method. The authors note that “We recommend using image data whenever possible to get the most accurate predictions”, but we hypothesize the larger drop is also probably caused by the difference between European and Latin American Spanish. We also observed a drop in performance, as discussed in our paper, when comparing TWilBert (a model trained in European Spanish) with TwBETO (the model released in this paper).

Given these caveats, we decided to downplay the fact that the heterogeneous network model is closer to the final vote and added the disclaimer as recommended. This was done by removing the mention of this validation from the introduction, discussing this limitation in the “Robustness over time” subsection, and adding the disclaimer and reference to the reweighting approach in the limitation section.

C3. F1 scores are a function of the decision threshold. The optimal decision threshold varies based on the predicted scores distribution, hence it changes across datasets.

Recommendation: Authors should opt for metrics that are robust to the differences between domains, such as AUC or area under the precision-recall curve (AUPR).

Agreed, we included ROC AUC metric to all tables and analyzed it accordingly.

C4. The manuscript states in l. 94 that:

"Even though there exists work that has explored user-level stance detection on Twitter data [15], the proposed approach requires pairwise similarity calculations for all users of interest, which limits the scalability of the approach" Looking at Darwish et al. 2020, it seems that their algorithm depends on UMAP for dimensionality reduction and DBSCAN for clustering. Both algorithms have hyperparameters that allow them to scale by considering a smaller number of neighbors.

It is not clear that the algorithm cannot run on the present data. In fact, Darwish et al. have applied it to datasets of 1.8M, 2.4M and two datasets of 2.6M tweets, which are in the same scale as the datasets in the manuscript.

We have added additional clarification on this point; while Darwish et al. have datasets with millions of tweets, they only ran their approach on a maximum of 5,000 users (Table 2 in their paper). As the authors outline in the Finding Stance Clusters section, they calculate three pairwise cosine matrices for every user in their selected subset and run umap on their pairwise cosine similarity matrices. For the users in Chile alone, generating a single cosine similarity matrix for all of our users using their approach was set to take over 100 days. We added an additional sentence about the cosine similarity bottleneck in the Related Work section.

C5. Abstract refers to "users' social networks" which is a bit ambiguous. To distinguish from the platforms, authors should consider using the term 'egonets'.

Agreed, we updated the abstract accordingly and other references found in the main text that might lead to a similar confusion.

C6. To deal with class imbalance, authors use resampling, which tends to leave out some examples from the majority class. Have you consider using class weights in the loss function?

We did try weighting observations by the inverse of their class frequency and also other static resampling approaches, but found that dynamic resampling improved training stability as mentioned in the paper. We hypothesize that this is in part due to the dynamic sampling of n_max tweets for each user at the start of a batch, which implies that even repeated users have a good chance of including different tweets when added to the batch. At least 83% of the users in each training set had 15 or more tweets in the dataset. We added this discussion as it was not mentioned before.

Minor comments

l. 474 Undefined reference to Table 2.

Corrected broken reference.

REVIEWER 2 RESPONSES

Weakness: Some issues regarding the impact of each model component and the methodology to train and evaluate them must be clarified. First, the social network component and its emerging communities may alone answer the problem of user stance in favor or against protests. Second, the labeling process for training the model is superficially explained to make a self-contained manuscript.

On the first point, we agree that the network alone carries significant signal, and have added additional language highlighting that the TSPA baseline serves as a network baseline without any account features. TSPA’s performance is surprisingly strong even without user text attributes, although the User Transformer outperforms it in some countries (when considering the ROC AUC performance). We added language further emphasizing that point in the Methods and Results sections. We further expand this point in the responses below.

On the second point, we added further description of the labeling methodology. See responses to the data description comments for further details.

1 Introduction: Regarding the first contribution, I would recommend that the authors also mention the impact of each compartment of the proposed architecture on the stance-detection problem. As reported in the following comments, it is important to justify the construction of an architecture based on three compartments (tweet, user, and network layers), given its high computational cost.

We added more details justifying this architecture choice in the introduction, as it allows us to evaluate the effect of context on the in-sample performance and generalization (this is highlighted across the first 3 contributions). Other details on how this effect is manifested are addressed in the points discussed below and in the original results section.

2. Related Work: I recommend explaining specific technical terms such as “hand-labeled” and “hand-crafted” to make the manuscript clear for readers from broad research areas.

Added further description to clarify the aforementioned terms.

3. Data Description

Authors must provide further information about the adopted methodology to construct a ground truth dataset of labeled comments and evaluate the proposed framework. First of all, the weak-labeling step of the methodology must be explained in order to produce a self-contained manuscript. Questions that are raised from the current data description are:

- How is the “endorsement of hand-labeled political figures” leveraged?

The methodology relies on the hypothesis that users are more likely to tweet (or retweet) stance-tags (hashtag campaigns with well-defined stances towards each government and that occur at the end of a tweet) or political figures that are aligned with their stances during these events. For this reason, stance labels are assigned to a user if the percentage of tweets with a consistent stance-tag or retweets from political figures with a consistent stance is above a given threshold (the authors use 90% as a threshold based on a labeled validation set). A final stance label is assigned to a user if the stance obtained by both signals (usage of hashtags and retweet of political figures) is consistent. This discussion was added to the paper.

- Which hashtag campaigns have well-defined stances toward each government? I recommend providing a table with prominent examples.

Added a table with prominent examples.

- Which polarized communities are observed according to user partitions from the constructed labels? It appears a network of users supports the weak-labeling methodology, but no information is available on how this network and connections between users are constructed.

The authors validate the quality of the weakly-annotated labels based on a hand-labeled sample of users and by showing that the constructed labels partition the users in communities that are polarized in their language and news-sharing behavior in a way consistent with the ideological underpinnings of each protest. These communities are not explicitly defined, but are shown to use semantically consistent text and expressions by leveraging an established machine translation algorithm. They show that terms related to left-leaning ideologies in one community tend to be discussed in similar contexts as right-leaning terms (e.g., Socialism mistranslates to Fascism); terms related to law and order in one group are discussed in a similar context as the other discusses oppression, or that opposition leaders are discussed in similar contexts as government representatives. This discussion was added with examples of how this polarization is manifested in the data description section.

The observation of polarized communities raises the issue of whether the users’ network and its emerging communities alone answer the problem of user stance in favor or against protests. What is the relevance (i.e., the impact) of the users’ text content for detecting their instances?

This issue should be addressed in the analyses, and the impact of each component on the stance problem in the context should be clarified.

Added a brief discussion of this objective in the data section and explore it in more detail in the results section as described in the responses below.

Why the weak-labelling is not used for the Chilean Referendum dataset?

As discussed in the “The 2020 Chilean Plebiscite 224” subsection, we applied the same weak-labeling methodology, based on two stance signals, to construct the referendum dataset. This section also provides further details on the weak-labeling methodology used for the construction of the protest dataset, addressing in part the previous comment and making the paper more self-contained (given that we follow the same algorithm).

However, as mentioned, the difference lies in that instead of using hand-labeled Political Figures, as was done to construct the weak-labels for the South American protests, we opted to identify the hashtags used in the user description that were explicitly rejecting or approving the referendum in first person. This simplified the labeling procedure given that we labeled other stance tags for the referendum. We found that the usage of stance tags in the descriptions as a replacement signal also resulted in meaningful polarized stance partitions based on the same language metrics used to validate the protest weak-labels (we did not explore polarization in news-sharing behavior during the referendum as it escaped the scope of this work). This discussion was added to the paper to clarify this topic.

4. Materials and Methods

Some notes and recommendations for better comprehension of Fig1:

- Include input arrows for each layer to indicate where to start reading each layer, mainly the first and second ones, which have more details.

Added a white arrow to denote the first input to each component. Other than this, the figure already included arrows that showed the flow through the architecture. However, if the white arrow for the last component adds too much clutt

---

## [Decision Letter · Decision Letter 1]

11 Mar 2025

PONE-D-24-21648R1Social context in political stance detection: Impact and extrapolationPLOS ONE

Dear Dr. Villa-Cox,

Thank you for submitting your manuscript to PLOS ONE. After careful consideration, we feel that it has merit but does not fully meet PLOS ONE’s publication criteria as it currently stands. Therefore, we invite you to submit a revised version of the manuscript that addresses the points raised during the review process.

We look forward to receiving your revised manuscript.

Kind regards,

Carlos Henrique Gomes Ferreira, Ph.D.

Academic Editor

PLOS ONE

Journal Requirements:

Reviewers' comments:

Reviewer's Responses to Questions

**Comments to the Author**

1. If the authors have adequately addressed your comments raised in a previous round of review and you feel that this manuscript is now acceptable for publication, you may indicate that here to bypass the “Comments to the Author” section, enter your conflict of interest statement in the “Confidential to Editor” section, and submit your "Accept" recommendation.

Reviewer #1: All comments have been addressed

Reviewer #2: All comments have been addressed

2. Is the manuscript technically sound, and do the data support the conclusions?

Reviewer #1: Yes

Reviewer #2: Yes

3. Has the statistical analysis been performed appropriately and rigorously? 

Reviewer #1: Yes

Reviewer #2: Yes

4. Have the authors made all data underlying the findings in their manuscript fully available?

Reviewer #1: Yes

Reviewer #2: Yes

5. Is the manuscript presented in an intelligible fashion and written in standard English?

Reviewer #1: Yes

Reviewer #2: Yes

6. Review Comments to the Author

Reviewer #1: I thank the authors for the detailed responses to my comments and suggestions. I have carefully read the answers to both reviews and changes made to the manuscript.

In particular, I appreciate their efforts in attempting to run demographic inference on their large dataset of tweet samples to determine whether the sample could be considered representative of Chile's population. I acknowledge that the changes to X/Twitter's API have hindered research efforts based on the platform's data and agree with the authors' decision of downplaying the proximity of the model's output to the final vote results as a form of validation. In addition, I appreciate the authors' efforts of making the data and code available for reproducibility purposes.

(New comment) One new issue came up after the inclusion of ROC-AUC results -- the statement "Protests in Bolivia had 506 opposing ideological motivations in comparison with the motivations observed in the 507 other countries," coupled with the shockingly low AUC results (specifically, all of them much lower than 0.5 ['random']) in Bolivia suggests a simple fix: to swap the positive and negative labels. High AUC values would actually lead to an opposing finding: that the results can generalize well across countries provided that there is a suitable mapping for the labels.

Recommendation: either the authors swap the labels or explain why this is not the right choice in their context.

Reviewer #2: I appreciate the authors' effort in preparing the manuscript after the first round of peer review. I believe the authors improved the quality of their manuscript compared to the previous version.

Below are some recommendations to be considered in the final version of the paper if it is accepted for publication in the journal.

Regarding confidence intervals in tables, it is not easy to see the best-performing model. Thus, the best-performing model should be bold for easier reading.

Show clearly the measure used to define the best-performing model. Is the measure F1 or ROC AUC?

Besides, I recommend clearly showing how the confidence interval was calculated. If it was based on mean values, why not show the mean plus-minus its error for the used confidence interval instead of percentiles? It is important to clarify these points in some paragraphs of the results section.

7. PLOS authors have the option to publish the peer review history of their article (what does this mean?). If published, this will include your full peer review and any attached files.

Reviewer #1: **Yes: **Fabricio Murai

Reviewer #2: No

---

## [Author Response · Author response to Decision Letter 2]

16 Mar 2025

Dear Reviewers and Editor,

We appreciate the consideration given to our response and thank you for the further feedback. In what follows we address the minor revisions posited in this new round of reviews.

Journal Requirements

We reviewed our reference list, and did not find any retracted references. However, while doing so we did find several instances of references that listed the arXiv preprint, when a peered review version was available. We proceeded to update said references and, as requested, list the changes made (The checkmark signals de updated version included in the new submission).

Updated References:

o GlobeNewswire. $6.78 Billion Public Opinion and Election Polling Global Market to 2030 - Identify Growth Segments for Investment; 2021. Available from: https://www.proquest.com/wire-feeds/6-78-billion-public-opinion-election-polling/docview/2555898927/se-2?accountid=9902.

+ GlobeNewswire. $6.78 Billion Public Opinion and Election Polling Global Market to 2030 - Identify Growth Segments for Investment; 2021. Available from: https://www.globenewswire.com/news-release/2021/07/29/2271041/28124/en/6-78-Billion-Public-Opinion-and-Election-Polling-Global-Market-to-2030-Idhtml.

o Kochkina E, Liakata M, Zubiaga A. All-in-one: Multi-task learning for rumour verification. arXiv preprint arXiv:180603713. 2018;.

+ Kochkina, E., Liakata, M., & Zubiaga, A. (2018, August). All-in-one: Multi-task Learning for Rumour Verification. In Proceedings of the 27th nternational Conference on Computational Linguistics (pp. 3402-3413).

o Xu C, Paris C, Nepal S, Sparks R. Cross-target stance classification with self-attention networks. arXiv preprint arXiv:180506593. 2018;.

+ Xu, C., Paris, C., Nepal, S., & Sparks, R. (2018, July). Cross-Target Stance Classification with Self-Attention Networks. In Proceedings of the 56th Annual Meeting of the Association for Computational Linguistics (Volume : Short Papers) (pp. 778-783).

o Mohtarami M, Glass J, Nakov P. Contrastive language adaptation for cross-lingual stance detection. arXiv preprint arXiv:191002076. 2019;

+ Mohtarami, M., Glass, J., & Nakov, P. (2019, November). Contrastive Language Adaptation for Cross-Lingual Stance Detection. In Proceedings of the 2019 Conference on Empirical Methods in Natural Language Processing and the 9th International Joint Conference on Natural Language Processing (EMNLP-IJCNLP) (pp. 4442-4452).

o Khiabani PJ, Zubiaga A. Few-shot Learning for Cross-Target Stance Detection by Aggregating Multimodal Embeddings. arXiv preprint arXiv:230104535. 2023;.

+ Khiabani, P. J., & Zubiaga, A. (2023). Few-shot learning for cross-target stance detection by aggregating multimodal embeddings. IEEE Transactions on Computational Social Systems, 11(2), 2081-2090.

o Smith SL, Turban DH, Hamblin S, Hammerla NY. Offline bilingual word vectors, orthogonal transformations and the inverted softmax. arXiv preprint arXiv:170203859. 2017;.

+ Smith, S. L., Turban, D. H., Hamblin, S., & Hammerla, N. Y. (2017, February). Offline bilingual word vectors, orthogonal transformations and the inverted softmax. In International Conference on Learning Representations.

o Wolf T, Debut L, Sanh V, Chaumond J, Delangue C, Moi A, et al. Huggingface’s transformers: State-of-the-art natural language processing. arXiv preprint

arXiv:191003771. 2019;

+ Wolf, T., Debut, L., Sanh, V., Chaumond, J., Delangue, C., Moi, A., ... & Rush, A. M. (2020, October). Transformers: State-of-the-art natural language processing. In Proceedings of the 2020 conference on empirical methods in natural language processing: system demonstrations (pp. 38-45).

o Bird S, Klein E, Loper E. NLTK book; 2009.

+ Bird, S., Klein, E., & Loper, E. (2009). Natural language processing with Python: analyzing text with the natural language toolkit. " O'Reilly Media, Inc.".

REVIEWER 1 RESPONSE

1. One new issue came up after the inclusion of ROC-AUC results -- the statement "Protests in Bolivia had 506 opposing ideological motivations in comparison with the motivations observed in the 507 other countries," coupled with the shockingly low AUC results (specifically, all of them much lower than 0.5 ['random']) in Bolivia suggests a simple fix: to swap the positive and negative labels. High AUC values would actually lead to an opposing finding: that the results can generalize well across countries provided that there is a suitable mapping for the labels.

Recommendation: either the authors swap the labels or explain why this is not the right choice in their context

We agree with the observation that we could leverage the low AUC results for the Bolivian case and our domain knowledge of the different ideological motivations behind the Bolivian protests; to maintain the high generalization capacities of the heterogeneous model (as mentioned in the paper, we take this approach when applying the Chilean classifier to the referendum data). However, we chose not to do so for the cross-country robustness analysis as the performance of the different classifiers (trained in other countries) when applied to the Bolivian case allows us to elaborate in the following interesting discussion (taken from line 520 to 525):

“As shown, the Bolivian case serves as an adversarial setting for classifiers trained in other countries, which suggests that, when applied to this country, the semantic features leveraged by the classifiers are operating on an ideological dimension. This is consistent with results presented in villa2022linguistic, where the authors found that language polarization remained along ideological lines when comparing protests of any of the three countries with Bolivia. This was not the case when comparing protests of the other three countries.”

And also to highlight (taken from line 547 to 552):

“ … the importance of domain knowledge when applying these models to different domains. If we leverage the knowledge of the opposed motivations for the Bolivian protests by inverting the heterogeneous model's predictions, its performance would be in line with what was observed for the other countries. We take the approach of inverting the predictions when we report the results of the 2020 Chilean referendum..”

We believe that this discussion sufficiently addressed the correct observation made in the review, and did not include additional modifications. However, if further clarification is needed or if this discussion should be addressed before in the section, we would be happy to revise it.

REVIEWER 2 RESPONSES

1. Regarding confidence intervals in tables, it is not easy to see the best-performing model. Thus, the best-performing model should be bold for easier reading

Show clearly the measure used to define the best-performing model. Is the measure F1 or ROC AUC?

As suggested, the name of the best performing classifier is now bold if it obtained the best performance (or was statistically tied) in both F1 and ROC AUC. We do not favor one metric over the other, as both provide important information on model performance, and we discuss the relevant differences in the result section. We updated the relevant part of each table description which now reads:

“Model performance metrics correspond to the median of 1000 bootstrapped samples of the test set. For each metric, the classifiers whose performance is not statistically different, at a 95% confidence level, from the best-performing model are highlighted in bold. We also highlight in bold the name of the best-performing model if it obtains the best results (or is statistically tied) in all presented metrics. The corresponding confidence bands, computed from the bootstrapped empirical distribution of each metric, are included in the appendix.”

2. Besides, I recommend clearly showing how the confidence interval was calculated. If it was based on mean values, why not show the mean plus-minus its error for the used confidence interval instead of percentiles? It is important to clarify these points in some paragraphs of the results section.

We presented the median results, obtained from the bootstrapped empirical distribution, for each metric and its corresponding confidence band (either 95% for the main results or 99% for the robustness analysis). We chose not to present mean values plus-minus its error as that would be an approximation which was not necessary given that we had access to the empirical distribution. We also had to report the bands and not a plus-minus notation as the empirical distributions were not symmetrical (showed slight skewness). As suggested, we added the following discussion to better clarify these points in the results section:

“We report the median result of the bootstrapped empirical distribution for each metric and compute its corresponding confidence band to assess whether the differences are statistically significant (these are presented in the appendix).”

---

## [Decision Letter · Decision Letter 2]

30 Apr 2025

Social context in political stance detection: Impact and extrapolation

PONE-D-24-21648R2

Dear Dr. Villa-Cox,

We’re pleased to inform you that your manuscript has been judged scientifically suitable for publication and will be formally accepted for publication once it meets all outstanding technical requirements.

Kind regards,

Carlos Henrique Gomes Ferreira, Ph.D.

Academic Editor

PLOS ONE

Reviewers' comments:

Reviewer's Responses to Questions

**Comments to the Author**

1. If the authors have adequately addressed your comments raised in a previous round of review and you feel that this manuscript is now acceptable for publication, you may indicate that here to bypass the “Comments to the Author” section, enter your conflict of interest statement in the “Confidential to Editor” section, and submit your "Accept" recommendation.

Reviewer #1: All comments have been addressed

Reviewer #2: All comments have been addressed

2. Is the manuscript technically sound, and do the data support the conclusions?

Reviewer #1: Yes

Reviewer #2: Yes

3. Has the statistical analysis been performed appropriately and rigorously? 

Reviewer #1: Yes

Reviewer #2: Yes

4. Have the authors made all data underlying the findings in their manuscript fully available?

Reviewer #1: Yes

Reviewer #2: Yes

5. Is the manuscript presented in an intelligible fashion and written in standard English?

Reviewer #1: Yes

Reviewer #2: Yes

6. Review Comments to the Author

Reviewer #1: I thank the authors for carefully addressing my previous comments. I agree with the authors that the decision of not flipping the predictions for the Bolivian case make sense from a methodological standpoint and that the reason is also presented and well-justified in the current version of the manuscript.

Reviewer #2: I appreciate the authors' effort in preparing the manuscript after the second round of peer review. I believe the authors improved the quality of their manuscript compared to the previous version, and all the comments have been addressed.

7. PLOS authors have the option to publish the peer review history of their article (what does this mean?). If published, this will include your full peer review and any attached files.

Reviewer #1: **Yes: **Fabricio Murai

Reviewer #2: No

---

## [Editor Report · Acceptance letter]

PONE-D-24-21648R2

PLOS ONE

Dear Dr. Villa-Cox,

I'm pleased to inform you that your manuscript has been deemed suitable for publication in PLOS ONE. Congratulations! Your manuscript is now being handed over to our production team.

Kind regards,

on behalf of

Dr. Carlos Henrique Gomes Ferreira

Academic Editor

PLOS ONE